# Experiences managing pregnant hospital staff members using an active management policy—A qualitative study

Mette G. Backhausen[1]*, Mette Langeland Iversen[1], Margrethe Bordado Sköld[2‡], Thora G. Thomsen[3,4‡], Luise Moellenberg Begtrup[5‡]

1 Department of Gynecology and Obstetrics, Zealand University Hospital, Roskilde, Denmark, 2 The Department of Occupational and Social Medicine, Copenhagen University Hospital, Holbaek, Denmark, 3 Department of Otorhinolaryngology and Maxillofacial Surgery, Zealand University Hospital, Koege, Denmark, 4 Department of Regional Health Research, Faculty of Health Sciences, University of Southern Denmark, Odense, Denmark, 5 Department of Occupational and Environmental Medicine, Bispebjerg Hospital, Copenhagen, Denmark

☯ These authors contributed equally to this work.
‡ These authors also contributed equally to this work.
* mgb@regionsjaelland.dk

**Data Availability Statement:** All relevant data are within the paper and its Supporting Information files. Complete interview data cannot be made

## Abstract

### Background and objective

During pregnancy, absence from work increases significantly. Job adjustments have been shown to decrease absences; however, studies show only half of pregnant women who need job adjustments receive them. Little is known about the viewpoints of managers and possible challenges in the management of pregnant employees. The aim of this study was to investigate the experiences and considerations of managers in relation to managing pregnant hospital staff members and to describe the experiences of an active management policy for pregnant individuals.

### Methods

A qualitative study based on five focus group interviews was conducted at five public hospitals in Zealand, Denmark with participation of 19 hospital managers, from 17 different wards, representing six different medical specialties. The interviews took place from February to May 2019. Thematic analysis was used to analyze the data.

### Results

Four themes were identified: (1) The everyday management, (2) Managerial dilemmas, (3) Acknowledging the workplace culture, and (4) Dialogue as a means for the working relationship. The managers' experiences revolved around investing a lot of effort into the working relationship with pregnant staff members by adjusting job tasks and work schedules while balancing work tasks between all staff members. The dialogue was considered central in order to identify the needs of the individual staff member.

publicly available due to ethical and legal reasons. The Danish Data Protection Agency only authorized analysis and storage of data, not public availability. Upon request, a list of condensed meaning units or codes can be made available on request to Mette G Backhausen (email: mgb@regionsjaelland.dk) or Lia Lund, The Danish Data Protection Agency, Zealand University Hospital (email: lglu@regionsjaelland.d).

**Funding:** This study was funded by The Working Environment Research Fund The funders had no role in study design, data collection and analysis, decision to publish, or preparation of the manuscript. The following authors received salary from the funder (salary from the grand given by the funder); Mette Backhausen, Mette Langeland Iversen and Luise Møllenberg Begtrup.

**Competing interests:** The authors have declared that no competing interests exist.

## Conclusions

Overall, management dialogue constituted a central tool in order to identify the needs of the individual staff member. A proactive and open approach increased the chances of a fruitful dialogue. The individual staff member, the influence of the workplace culture, and the every-day management of the workplace all shaped the experiences of the managers. The concept of an active management policy for pregnant individuals was perceived to entail useful elements, but also as replicating what managers already did.

## Introduction

In Scandinavian countries, approximately 80% of women of reproductive age are employed outside the home [1]. During pregnancy, absence from work increases significantly [2, 3]. Data from a Danish report suggests that two out of three women are absent from work during pregnancy, and similar data exists from other Scandinavian countries [2, 4]. The absence represents a substantial socioeconomic burden [2] and causes challenges for workplaces that lack experienced staff and for colleagues that either have to work understaffed or devote scarce resources to recruit and train new staff. Furthermore, studies have shown that absence from work affects pregnant women on both a personal and social level: they express feelings of guilt and frustration when unable to fulfil their work obligations and worry about the risk of social isolation [5–7].

In Denmark, public employees, including hospital staff members, have the right to maternity leave eight weeks before the due date. In cases of pregnancy-related sick leave, the staff member receives full pay and the workplace is compensated. Occupational exposures, i.e. lifting, shift work, and job requirements, are associated with increased risk of absence during pregnancy [8–10]. Furthermore, existing evidence shows that the majority of sickness absences are related to normal pregnancy discomforts such as fatigue, nausea, low back pain, pelvic pain and Braxton Hicks contractions [11, 12]. Job adjustments, defined as any changes made in working conditions in order to adapt to the needs of pregnant women e.g. changes in work schedule and in job tasks, have been shown to reduce work absences during pregnancy. However, studies show that job adjustments are only made for half of the pregnant employees who state a need for it [13–15]. This could indicate lack of focus from the employer, on the working environment among pregnant women. Qualitative studies based on interviews with working pregnant women find that the pregnant women lack awareness and consideration of their pregnancy at work [16, 17]. Managers were considered the most significant stressor at work, because of a negative attitude towards the occurrence of pregnancy [17, 18]. There is a lack of studies identifying these issues from a managerial point of view.

In a report from the Danish National Research Center for Welfare [3], company representatives state that by 2010, focus on pregnant women's working environments had increased. Despite the increased focus, rates of pregnancy-related sick leave remained high. The company representatives were unable to pinpoint the main reason, but mentioned workloads and lack of opportunity for job adjustment as possible explanations [3]. To our knowledge, no other study has investigated the managers' perspectives concerning these issues, which is important if initiatives to decrease sickness absences are to be implemented. Further in-depth insight into the mechanisms between managers and their pregnant staff members is needed to be able to improve the working environment for pregnant women and prevent absence during pregnancy.

The aim of this study was hence to investigate the experiences and considerations of managers in relation to the management of pregnant hospital staff members and to describe experiences using an active management policy for pregnant individuals.

## Methods

### Study setting and design

The study was performed at Zealand University Hospital as a continuation of a randomized controlled trial (RCT). The RCT investigated if a seminar for hospital managers focusing on how to have an active management approach to pregnant staff members would increase well-being and reduce absences among pregnant staff members [19]. This active policy entailed managers implementing structured conversations early on and throughout an employee's pregnancy, at least three times (See S3 File). Five hospitals in the Capital and Zealand region, Denmark, participated in the RCT study.

For the current study, we chose to conduct focus group interviews, since this method is suitable to explore shared experiences and attitudes towards a mutual topic of interest [20–22]. Participants were managers who participated in the RCT seminars and had experience managing pregnant staff members. Reporting on the study was performed in accordance with COREQ, the consolidated criteria for reporting qualitative research [22]. The research group consisted of five female researchers within three professions: midwifery (MGB has a Ph.D, MLI has an MSc), nursing (TGT has a Ph.D) and doctors of occupational medicine (MBS and LMB. LMB has a Ph.D). All authors had clinical experience and experiences in research, including qualitative (MGB, MLI, MBS, TGT) and quantitative methods (MGB, MLI, MBS, LMB).

With the aim of the study, we seek to understand the basis on which managers make their decisions, when they interact with pregnant employees. We acknowledge that we are part of the analysis and that we consciously use all members of the research team supplement and contest each other's statements during the analysis [23].

### Procedures and sample

Eligible participants for the current study were the 86 hospital managers with direct staff supervision responsibility on a daily basis, and who participated in a seminar as part of the RCT study. An invitation was emailed to all eligible participants with information on the focus group interview study. After 2 weeks, the managers were contacted by telephone and asked if they wanted to participate. To ensure information rich cases [20], the managers were also asked if they had any experience with pregnant staff members in the one year study period. Five focus groups interviews were planned, one in each of the five participating hospitals. This was done to ensure that it would be convenient for the managers to participate. Managers interested in participating were invited to one of the scheduled focus group interviews. Of the 86 eligible participants, 26 were scheduled for a focus group interview. Seven cancelled before the interview, mainly due to lack of time, leaving a final sample of 19 managers. Five focus group interviews with 3–6 participants each were conducted (N = 19).

The first focus group interview was performed as a pilot interview and as no changes to the interview guide were necessary, the pilot interview was included in the final analysis.

### Interviews

The focus group interviews took place at the end of the study period, one year after participation in the RCT seminar, from February to May 2019, at five Danish hospitals in the Capital

and Zealand Regions. The interviews were carried out by the first and second author (MGB and MLI), who served alternately as either moderator or observer. Field notes were taken by the observer and discussed with the moderator at the end of the interview. The interviews lasted between 55 minutes and 1 hour 28 minutes. Based on the study aim, an interview guide was developed covering three topics (See S2 File). The topics comprised: 1) the managers' general experiences with pregnant staff members, 2) specific experiences working with an active management policy for pregnant individuals, and 3) their experiences with prevention of sick leave and potential initiatives to improve the working relationship with pregnant staff members. To allow the participants to elaborate freely on their own experiences, a funnel-based interview strategy was undertaken [21, 24], with each interview starting with a broad question and ending with more specific questions. Before each interview, the participants were informed about the moderators' profession (midwives with advanced education in health research) and that the moderators had no connection to the RCT study. In addition, they were asked to complete a short questionnaire including: age, gender, profession, years of experience in a managing position, number of pregnant staff members managed in the study period, and whether their pregnant staff members worked daytime or shifts.

## Data analysis

The focus group interviews were audio recorded in full and transcribed verbatim by secretaries with no connection to the study. NVivo 12 was used to organize the data. Data was analyzed using inductive thematic analysis as described by Braun and Clarke [25]. This analysis provides a method for identifying, analyzing and reporting patterns within data. It is a non-linear six-phase process where the analyst can move back and forth between phases as needed. First, the interviews were read through to familiarize the researchers with the depth and breadth of the content. Second, initial codes were produced based on meaningful text regarding the study aim. The coding was performed by the first two authors (MGB and MLI). Third, themes were identified and a theme map drafted. All authors met and discussed potential themes until consensus was reached on the preliminary themes. Fourth, the preliminary themes were reviewed by the first two authors (MGB and MLI), which involved going back and forth from the interview transcripts, codes and themes to refine the themes. Themes and sub-themes were identified based on multiple meaningful units. All authors met again, discussed, and finalized the themes and sub-themes. Fifth, the final themes were named and the refinement, description and structure discussed until consensus was reached. Finally, the results were elaborated, including quotes to exemplify how the final themes were identified. No informant checking was performed. According to Saunders et al. [26], data saturation in qualitative research should be operationalized in a way that is consistent with the analytic framework adopted. In this study, we have chosen an induction thematic analysis, which means that data saturation relates to the emergence of new themes. Data saturation was thus obtained when no new themes emerged.

## Ethics

All participants received written and oral information of the study purpose. Written informed consent was obtained before each interview. Participants were informed that all data would be treated confidentially and anonymously, and that they were free to withdraw from the study at any time. The study was approved by The Danish Data Protection Agency (no. REG-057-2018). According to Danish law, ethical approval is not required for non-invasive studies, including qualitative interview studies. Modified data is available upon request.

## Results

This study included 19 hospital managers from five different hospitals. The managers were employed at 17 different wards, both in- and outpatient, representing medical and surgical departments, pediatrics, gynecology and obstetrics, emergency medicine, and neurology departments. The participants had a median age of 49.6 years and their managerial experience varied from one to 34 years (Table 1).

The interviews resulted in four themes and 10 sub-themes, as shown in Table 2.

### The everyday management

This theme described the reflections of managers' everyday management of their department and included thoughts of job adjustments in order to keep pregnant staff members working, while balancing work tasks and not overloading the remaining staff. From the discussions, a diversity of the managers and their approaches toward pregnant staff members emerged; some managers expected the pregnant staff member to do their job as per usual and approach the manager in case of work-related challenges. Others expressed making decisions on behalf of their staff member; others involved the pregnant staff member in decisions, and to a certain extent let the staff member decide on the organization of their work.

**Willingness and effort put into job adjustments.** The managers recognized the pregnant staff members' need for adjustments to the work and work schedule during pregnancy, e.g. by reassigning them from night shift to day or evening shift, in order to retain them at work. Although their approach to achieve this varied, the managers expressed a general willingness to go far in order to find a suitable solution for both the pregnant woman and the department.

*"You stay ahead of things and align your expectations and get that signal sent; you know what, I will do everything possible so that you can stay at work, there are plenty of options"* (FG 2)

**Table 1. Characteristics of the study participants (n = 19).**

| Characteristics* | Number |
|---|---|
| **Age (years)** | |
| Median (min-max) | **49.6** (37–62) |
| **Profession** | |
| Nurse | **14** |
| Midwife | **4** |
| Other | **1** |
| **Years of experience in a managing position** | |
| Average | **12.6** |
| 1–5 years | **5** |
| 6–10 years | **5** |
| >10 years | **9** |
| **Number of pregnant staff members within the last year for each manager** | |
| 1–5 pregnant staff members | **11** |
| ≥6 pregnant staff members | **8** |
| **Work schedules in managers unit** | |
| Only Daytime | **3** |
| Shift Work | **16** |

*Both men and women participated in the study, but the majority were women. To ensure anonymity, gender will therefore not appear in the table.

**Table 2. Themes and sub-themes identified in the analysis.**

| Themes | Sub-themes |
| --- | --- |
| The everyday management | • Willingness and effort put into job adjustments |
| | • Balancing work tasks |
| Managerial dilemmas | • Predicting sick leave |
| | • An increased need for security |
| | • Changing priorities |
| Acknowledging the workplace culture | • Sharing before caring |
| | • The significance of the colleagues |
| The dialogue as a means for the working relationship | • The fruitful dialogue |
| | • The challenging dialogue |
| | • The absence of dialogue |

However, some also expressed a concern about being overprotective and labelling the pregnant staff members as having special requirements and needing job adjustments, which may not have been the case, and thereby making them seem sicker unnecessarily. Having participated in the RCT led the managers to reflect upon their approach to pregnant staff members. While some expressed they used the structured approach presented at the seminar, others stated that the intervention had no influence on the way they approached pregnant staff members.

*"Before, it was a little more, it could be while you were having a coffee. Now it is very formal and structured. That could very well be a good thing, because it could well be that the pregnant staff member. . .looks uncomplicated and just goes about their work. It may well be that if you actually asked them, they might need a formal meeting." (FG 4)*

**Balancing the work tasks.** The consideration of trying not to overburden the rest of the staff was described as a major concern by the managers as they argued for the importance of maintaining the balance of work tasks and consideration for pregnant staff members. This balance was often mentioned as a necessary condition before initiating any job adjustments for pregnant staff members.

*"But I also think it's important that there's some sort of plan in place for the other members of staff, as it also has to come together for them, because that's the balance we're finding at the moment whilst we're short on resources, this thing where by sparing some people, we're doing it at the expense of others." (FG 5)*

However, the general perception among managers was that any department should be able to accommodate pregnant staff members and, if necessary, find solutions to keep them at work for as long as possible. In regards to this, managers emphasized the importance of reacting positively when a staff member first announces her pregnancy and keeping to themselves concerns about staff shortages and work planning.

*"I think that it is quite fundamental for the remaining time, the way you meet them in that particular second they come in and sit down on the chair and tell you that there's something they'd like to say." (FG 1)*

It was mentioned as challenging to balance job tasks if a department had several pregnant staff members or other staff members on long-term sick leave. It was therefore considered important not to enter into too many special agreements with e.g. pregnant staff members without considering the entire staff group.

## Managerial dilemmas

Discussions during the interviews reflected how the managers experienced a shift in approaches and attitudes toward work among some pregnant staff members and how this affected their work capacity, their ability to cope with work, and days spent on sick leave. This shift sometimes resulted in a change in the working relationship between manager and staff member.

**Predicting sick leave.** The managers described that often they were able to predict which staff members would go on sick leave during pregnancy. This was based on sick days taken prior to pregnancy, as well as personal characteristics such as age, parity, robustness and industriousness.

*"In my experience you can keep a great many working for good long while, but I can pretty much spot them beforehand, I can predict who's going to call in sick and who won't, because this here also corresponds with their general attitude towards illness and sick leave." (FG 1)*

In the managers' experience, industrious pregnant staff members are generally easily motivated to continue working during pregnancy, while pregnant staff members who are less industrious often use pregnancy as an excuse to take sick leave. While some managers gave great importance to previous sickness leave, others described how they tried to avoid stereotyping the pregnant women and instead saw them as individuals.

**An increased need for security.** An increased level of anxiety among some pregnant staff members was mentioned. This was seen in relation to specific work tasks such as handling patients with infectious diseases, busyness and in the administration of various drugs. Having these concerns led to an increased level of insecurity, which was according to the managers, also affected by family and friends who voiced myths about how work could have a negative effect on their pregnancy and baby, and thereby caused anxiety in the pregnant staff member. In these situations, the managers experienced that pregnant staff members often expected managers to have knowledge in relation to pregnancy and risk related to specific work tasks. Managers who could not meet this expectation contributed to a sense of mistrust, where supervisees questioned whether the managers had the necessary and most up-to-date knowledge of pregnancy and working conditions; this in turn caused supervisees to question whether managers would take necessary precautions on their behalf.

*"This is important for the pregnant women, that there's documentation for what we say is correct, that's it isn't just because we want them to come to work." (FG 3)*

A booklet provided at the seminar was considered useful because it contained necessary knowledge and clarified existing evidence regarding pregnant staff members and the work environment. To further increase a sense of security for these women, some managers suggested extra precautions, even though they weren't required, e.g. restricting pregnant staff members from working with patients with infectious diseases.

**Changing priorities.** Managers also mentioned a change in priorities among some pregnant staff members, who shifted their focus from work toward family and other private matters. A decreased physical capacity became another area of concern for some pregnant staff

members in their everyday work, disturbing the work-life balance and making work more challenging. The managers also witnessed changes in mental state, which could sometimes complicate the working relationship.

> *"This is about their work mentality. Do I want to be a part of this still, or am I so engrossed by the fact that I´m about to become a mother, and can I cope or work with the symptoms my body now gives me because of it? Some can and some can't, they're simply just so mentally pregnant all over, and I find them really hard to reach." (FG 2)*

## Acknowledging the workplace culture

Workplace culture was described by managers as dynamic, under the influence of everyday management, the managers' own approach, other colleagues, and the diversity of the pregnant staff members. The dynamics of the workplace culture were considered important for the potential influence on the working relationship with pregnant staff members, in both positive and negative ways.

**Sharing before caring.** Openness, in general, was considered an important element by the managers, having the potential to positively affect the workplace culture. The managers considered the pregnant staff members' openness toward their colleagues with regards to their physical limitations and challenges as an important element in planning work tasks and individual agreements. This was perceived as an improvement to the work environment for all staff members and encouraged the other staff members to help pregnant staff members when necessary. The managers considered their own openness towards the pregnant staff members as positively influencing the working relationship by emphasizing the importance of the individual staff member's contribution to the department.

> *"The important thing is what we each share out loud, what we can and cannot do, so colleagues know that they're pregnant and can help them." (FG 2)*

**The significance of colleagues.** The general working environment and culture in a department was considered a central factor. If pregnancy was discussed among colleagues as a natural process, it could help to demystify pregnancy and unnecessary work restrictions. If, on the other hand, colleagues expressed concerns about pregnancy, the managers experienced that a negative culture was created, and pregnant staff members were more likely to take sick leave.

> *"…if there is a culture that dictates that when you are pregnant, you have to take sick leave, then I believe the attitude will spread, and more will eventually become sick listed" (FG 3)*

In order to positively influence the workplace culture, the managers used the booklet from the seminar to illustrate facts concerning pregnancy and work and to focus on the work environment during pregnancy. The managers believed that formerly pregnant staff members also acted as role models and affected whether a current pregnant staff member would stay at work until maternity leave or not.

> *"There's just been a couple who've gone, like, who've gone a whole pregnancy until, well, almost until maternity leave, and I think there's also a bit of prestige in that, right?" (FG 4)*

On the other hand, a negative attitude among pregnant staff members was considered contagious and could potentially affect the overall well-being of the remaining staff members in a

negative way. This caused the managers to argue that pregnant staff members should not remain at work at all costs.

## The dialogue as a means for the working relationship

Managers agreed on the importance of having conversations with pregnant staff members as this was seen as a way to ensure a good working relationship, and thus such conversations were prioritized in busy work schedules. The relationship was considered successful if the pregnant staff member continued working until maternity leave was due. For some managers, organizing such conversations was described as a time-consuming task which didn't always pay off in the end. The conversations contained some of the challenges described above and had certain characteristics as described below.

**The fruitful dialogue.**    A conversation was described positively when the manager and the pregnant staff member wanted the same goal, when there was agreement on which tasks to perform, and when these tasks were beneficial for both the staff member and the workplace. It was considered particularly positive when the pregnant staff member suggested specific tasks that seemed feasible to her, and when she was honest about limitations while taking co-responsibility in finding solutions. This led to greater goodwill for making adjustments to work tasks by the managers.

> "*When it all comes together. When the staff member feels she's been heard, and we also feel it's working a bit to our advantage too, without it becoming way too complicated.*" (FG 5)

**The challenging dialogue.**    These conversations could also be labeled as challenging by the managers. This was the case if the pregnant staff member was limited in completing common work tasks but did not discuss these limitations with the manager. This led managers to perceive the staff member as being less honest or too private about challenges, which made adjustments of work tasks difficult and would lead to sick leave, especially during weekends and long shifts. It was perceived as challenging when the pregnant staff member questioned the managers' knowledge about precautions during pregnancy and ultimately went elsewhere to seek knowledge on their own.

> "*She simply didn't trust me during the conversations I've had with her about which precautions we're taking, what was needed, and so it went completely wrong.*" (FG 3)

The managers often experienced that if the conversations were challenging, it led the pregnant staff member to seek advice from e.g. at their general practitioner, having involved the manager only partly in the problem. The involvement of the general practitioner often caused the pregnant staff member to be placed on sick leave. This was described as frustrating since managers felt partly excluded from the possibility of finding a solution.

**The absence of dialogue.**    At times, these management conversation were also be perceived as failures. This was the case if the staff member didn't present her problems at all, and didn't engage in a dialogue with the manager about possible solutions. As a result, the pregnant staff member was often placed on sick leave by her general practitioner even before the manager became involved. This made the managers feel powerless and reluctant to find possible solutions that might retain the staff member at work, and therefore this often resulted in longer sick leaves.

> "*By just ringing in and saying I'm going off sick, I'll just send in my doctor's sick note. Then you're just not in a dialogue.*" (FG 4)

It was emphasized that too often the general practitioners placed the pregnant staff members on sick leave without considering other possible solutions. The managers called for a more systematic approach in evaluating degrees of ability or limitations among pregnant staff members in relation to their work.

## Discussion

This study explored managers' experiences and considerations in the management of pregnant hospital staff members and the experiences obtained using an active management policy for pregnant workers. Overall, the study findings reflect a number of interrelated dimensions involved when discussing work and pregnancy from the perspective of the managers. Four categories were identified during the analysis. The first category covered the effort managers invest into the working relationship with pregnant staff members, such as adjusting tasks and work schedules while balancing work tasks between all staff members. Secondly, the study identified the challenges the managers faced when insecurity or a change in priorities occurred for some pregnant staff members. Thirdly, the study showed how important openness and communication is in order to create a positive work environment. The final category reflected the diverse and challenging role of communication between manager and pregnant staff members, leading to management dialogues with varying positive and negative outcomes.

The literature on work situations leading to sick leave among pregnant women is very limited. An interesting finding in our study was that the managers expressed that they devoted a lot of effort into job adjustments, including changing work schedules. This is in contrast with existing evidence [13–15]. A Danish study that interviewed new mothers found a lack of support from managers at work during pregnancy and that they often felt caught in a principle of either-or, meaning that job adjustments were needed but rarely given [7]. This illustrates how differently communication and support during pregnancy can be viewed depending on the perspective. It further underlines the importance of a good management dialogue and alignment of expectations. Management dialogue itself was a central topic throughout the focus group interviews and was considered fruitful if both parties contributed constructively, particularly if the pregnant staff member showed an interest in staying at work and brought suggestions and solutions into the dialogue. The importance of dialogue as a tool in the management of pregnant employees is supported by two other studies among non-pregnant individuals. A Canadian study showed that improved insight in the worker's perspective may improve working relations between workers returning to work after a longer period of sick leave and their manager [27]. A Dutch study demonstrated that frequent contacts with a supervisor were associated with favorable return to work rates [28]. The results of the studies may not be transferable to a population consisting strictly of pregnant individuals.

Another important finding was that the managers described some of the pregnant staff members as being anxious and worried about their pregnancy, which represented a challenge for the working relationship and sometimes led to sick leave. The managers further argued that they could predict the ones taking sick leave during pregnancy. A recent study showed that pregnant women often feel guilty due to their reduced capacity for work, saw themselves as a burden to their colleagues, and were conflicted if they needed to be on sick leave [7]. This indicates that the decision to take sick leave may not be as easy as some managers think, and underlines a complexity faced by pregnant staff members, on which managers sometimes may fail to understand.

Another interesting result was that the managers felt that health professionals were of little help in keeping pregnant staff members at work and that general practitioners often signed the pregnant staff members off sick, without considering other solutions. Managers often do not

have the health professional credentials to assess whether sick leave is necessary or not, however they may have knowledge to overcome unnecessary worry and thereby prevent unnecessary sick leave. A future study could aim to investigate the role of health professionals and the effect of an approach with early detection of pregnant women at risk of taking sick leave, for example by involving doctors in occupational medicine.

The managers experienced that some women felt an increased anxiety regarding specific work tasks, e.g. handling patients with infectious diseases or administrating drugs. This anxiety was perceived as increasing the pregnant staff members' expectations that managers would have specific knowledge about occupational risk factors during pregnancy. While some managers found that the booklet from the seminar provided this knowledge, others contacted the department of occupational and environmental medicine in order to gain more knowledge. This finding underlines the importance of having an early dialogue to address possible fears and worries, including a discussion of job tasks and possible job adjustments, in order to ensure well-being and a sense of security for pregnant staff members and thereby retaining them as workers. It also stresses the importance of the latest knowledge being available to all managers with responsibility for pregnant employees, in order to support them in the management. According to hygiene specialists, all hospital staff members are encouraged to comply with Hospital standards regarding hygienic precautions and use protective equipment as recommended in each patient category. This is considered sufficient to avoid infectious diseases.

The active management policy of pregnant individuals was developed to support the dialogue early on and throughout pregnancy. However, in the present study, the managers described different approaches to and opinions about the concept, showing that the concept was not fully implemented and that managers primarily organize their work based on their own experiences. A Dutch study investigated if a 'conversation roadmap' could enhance cooperation between employees on sick leave and their supervisors in order to support earlier returns to work in a banking organization, but found that it was only used to a limited extent [29]. This suggests that too structural and strict changes may be difficult to implement at a managing level.

As our results reflect, keeping pregnant staff members at work or helping them return to work after a period of sick leave is a complex process, affected by a range of factors. Existing literature from other fields of research integrates and investigates interrelated dimensions that together influence work ability and performance and may be of help in understanding the challenges faced by working pregnant women. The "Cancer and work" model [30] is an example of this. The model takes into consideration multiple dimensions that affect work performance: staff characteristics, symptoms, health and well-being, functions, work demands, and work environment [30]. This complexity is further described in relation to pregnancy in a narrative review [18] investigating implementation of the maternity protection legislation in industrialized countries and its expected and unexpected effects. The review stresses the importance of viewing maternity protections in a broader context, and its implementation to progress in the future [18]. These dimensions support our findings, as does conceptualizing pregnancy within a broader context, together with the challenges it may cause in a working situation.

## Strength and weaknesses

The credibility of the study was reflected in the participant sampling diversity in regard to age, profession, years of experience as manager and variation in medical specializations, allowing various perspectives on the research questions to flourish. Triangulation among the researchers, from three different professions, served to secure the trustworthiness of this study; during

the process of analysis, several meetings were held discussing themes and subthemes until consensus was reached [23].

The study was performed in a Danish hospital setting and the transferability [23] of our results may be limited to workplaces with similar working conditions, maternity protection legislation and organization of hospital services and should therefore be seen in this light. It is also worth considering that the managers who participated in this study have education in the health professions, which potentially could affect their approach to pregnant staff members and thereby the results.

Although interest in participating in the study was very positive among managers, it was challenging to carry out due to last minute cancellations on the day of the interview. This resulted in three focus groups with only three participants, although 6–10 participants are recommended for focus group interviews [21]. However, the same interview guide was applied and the discussions centered on the same topics as in the other focus groups. Furthermore, the participants in all the focus groups were encouraged to engage in discussions and freely express their points of view. The smaller number of participants in some of the focus groups may therefore not have affected the results considerably.

## Conclusions

Overall, managers' experiences with pregnant staff members are reflected in a number of inter-related dimensions. The dialogue was central in identifying the needs of individual staff members: a proactive and open approach increased the chances of a fruitful dialogue. The individual staff member, the influence of the workplace culture, and the everyday management of the work place all shaped the experiences of the managers. The concept of an active management policy for pregnant individuals was perceived as entailing useful elements, but also as replicating what mangers already did when working with pregnant staff members.

## Supporting information

**S1 File. Interview guide Danish version.**
(PDF)

**S2 File. Interview guide English version.**
(PDF)

**S3 File. Overview of active management policy and intervention.**
(PDF)

## Acknowledgments

The authors thank all the managers for participation in the focus group discussions and for sharing their experiences.

## Author Contributions

**Conceptualization:** Mette G. Backhausen, Mette Langeland Iversen, Margrethe Bordado Sköld, Thora G. Thomsen, Luise Moellenberg Begtrup.

**Data curation:** Mette G. Backhausen.

**Formal analysis:** Mette G. Backhausen, Mette Langeland Iversen, Margrethe Bordado Sköld, Thora G. Thomsen, Luise Moellenberg Begtrup.

**Funding acquisition:** Luise Moellenberg Begtrup.

**Investigation:** Mette G. Backhausen.

**Project administration:** Luise Moellenberg Begtrup.

**Supervision:** Thora G. Thomsen, Luise Moellenberg Begtrup.

**Writing – original draft:** Mette G. Backhausen, Mette Langeland Iversen.

**Writing – review & editing:** Margrethe Bordado Sköld, Thora G. Thomsen, Luise Moellenberg Begtrup.

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
