## [Decision Letter · Decision Letter 0]

13 Nov 2020

PONE-D-20-20150

Experiences managing pregnant hospital staff members using an active management policy – A qualitative study

PLOS ONE

Dear Dr. Backhausen,

Thank you for submitting your manuscript to PLOS ONE. After careful consideration, we feel that it has merit but does not fully meet PLOS ONE’s publication criteria as it currently stands. Therefore, we invite you to submit a revised version of the manuscript that addresses the points raised during the review process.

Please find the reviewer comments below. The reviewers have requested additional clarification on several points, particularly regarding the level of detail presented in the methods, in order to ensure that the study is reproducible by another researcher (PLOS ONE publication criterion #3). The reviewers have also provided suggestions for the framing of the discussions section, as well as other major and minor revision requests.

We look forward to receiving your revised manuscript.

Kind regards,

Hanna Landenmark

Associate Editor

PLOS ONE

Journal Requirements:

2. Please include a copy of the topic/interview guide used in the study, in both the original language and English, as Supporting Information, or include a citation if it has been published previously.

3. Thank you for stating the following financial disclosure. Please amend your Financial disclosure statement to declare sources of funding, or state that the authors received no specific funding:

Reviewers' comments:

Reviewer's Responses to Questions

**Comments to the Author**

1. Is the manuscript technically sound, and do the data support the conclusions?

Reviewer #1: Partly

Reviewer #2: Yes

Reviewer #3: Yes

2. Has the statistical analysis been performed appropriately and rigorously? 

Reviewer #1: N/A

Reviewer #2: N/A

Reviewer #3: Yes

3. Have the authors made all data underlying the findings in their manuscript fully available?

Reviewer #1: No

Reviewer #2: Yes

Reviewer #3: No

4. Is the manuscript presented in an intelligible fashion and written in standard English?

Reviewer #1: No

Reviewer #2: Yes

Reviewer #3: Yes

5. Review Comments to the Author

Reviewer #1: 1. The introduction section needs to clearly define what is meant by 'job adjustments' and 'active management policy' to lay the foundation for the manuscript.

2. The aim of the study as it reads is to obtain manager perception of sickness absence while in the introduction there is more reference to perspectives of pregnant staff. Statements such as studies 'show that job adjustments are only made for half of the pregnant employees who state a need for it. This could indicate lack of focus on the working environment among pregnant women' makes it confusing if pregnant staff are not seeking their entitlements or if the managers are denying it.

3. The methods section needs to be clarified further as in one instance it is said that there were 5 FGDs but there were 19 participants in total. Is it possible that some of the FGDs had 4 or less participants? why were there 5 FGDs in the first place? and were there any differences in who participated in which FGD (this is mentioned in limitations but need to be explained upfront in the methods section itself).

4. Given that the FGDs were held a year after the seminar participation of the managers, it is possible that all their views are after learning the correct approaches and can be a major limitation of the study.

5. The chosen themes are not self-explanatory and do not seem corroborating with the sub-themes. Themes are usually more crisp and punchier than the sub-themes. The articulation of themes need to be from the manager perspective - as it seems it is unclear if they are manager perspectives of staff perspectives.

6. There can be a better choice of quotes to align with the themes - it is important to mention if the sub-themes were coming from many participants or just one participant in one FGD.

7. Given that the study takes an inductive perspective, the themes need to be brought together to develop a coherent theory and this is not clear in the discussion.

8. Reflexivity and positionality of the authors need to be made more explicit in the methods section.

Reviewer #2: I congratulate the authors to an interesting paper. I have some comments.

Triangulation mostly describes involving using multiple sources of data in an investigation. The authors describe that they used triangulation meaning that different health care professionals/researchers were analyzing the data. However, this study could be referred to a mixed method study if the qualitative study was planned from the project’s start to complement the RCT. But, the published study protocol does not describe the qualitative study, eg population, data collection or the qualitative analysis.

I want more information about the intervention ie. the education of the leaders, which can increase the value of the study’ and make it reproducible.

Reviewer #3: ABSTRACT

It is unclear from reading the abstract by itself where this study took place. An additional sentence clarifying location and sample population would be useful.

DATA AVAILABILITY

Per my reading of the PLOS ONE guidelines, my understanding is that the Methods section should contain information regarding how to access the data used in this study - even if the full dataset is not publicly available.

PREDICTIONS

Even though this study is largely qualitative in nature, a set of predictions would be a good addition.

DISCUSSION

Much of this paper hinges on the idea of unnecessary sick leave (at least from the manager perspective) and how to prevent it. I would've liked to see a deeper discussion of pregnancy itself and how variable symptom prevalence/severity, incidence of medical complications, and real occupational threats factor in to the dynamic between pregnant individuals and the workplace. How does one determine if sick leave is necessary or not? I think this is especially critical, given the observation that medical practitioners appear to have a very different threshold for workplace risk than managers. I'd also like to know more regarding actual exposure risk in these work environments and to what degree women are right to be worried about possible infection (especially considering certain aspects of immune function are down-regulated during pregnancy, producing increased risk of viral infections). How many of these managers oversaw acute health settings (e.g. ICU) versus more chronic long-term care (e.g. med-surg), and does this factor in to employee anxiety regarding sick leave? Lines 171-175 give some idea of the dataset makeup, but little is said regarding possible differences between types of healthcare in the results/discussion.

LINE EDITS

line 72-73: The wording here is confusing. Do pregnant women themselves lack awareness, or do their managers/co-workers lack awareness? I assume you mean the latter, but restructuring this sentence would easily clear this up.

line 238: Unnecessary use of "e.g."

line 264: Labeling all concerns from friends and family as "myths" is probably too strong a statement. The way it is currently written does not clarify whether this was the chosen wording of the managers or of the authors.

6. PLOS authors have the option to publish the peer review history of their article (what does this mean?). If published, this will include your full peer review and any attached files.

Reviewer #1: No

Reviewer #2: No

Reviewer #3: No

---

## [Author Response · Author response to Decision Letter 0]

29 Jan 2021

We would like to thank you for your constructive and extensive review of our research article. We appreciate all the feedback and we are convinced that it has contributed to improve the manuscript significantly.

Reviewer #1, comment 1

The introduction section needs to clearly define what is meant by 'job adjustments' and 'active management policy' to lay the foundation for the manuscript.

Author’s response to the Reviewer #1, comment 1

Thank you for this comment. We agree that these two central elements should be explained further. 

In the introduction, we have defined job adjustment: 

Page 4, line 72 

changed from

Job adjustments have been shown to reduce work absences during pregnancy

To:

Job adjustments, defined as any changes made in working conditions in order to adapt to the needs of pregnant women e.g. changes in work schedule and in job tasks, have been shown to reduce work absences during pregnancy

To provide better insight to the concept “Active management policy” we have added supporting information with a schematic overview over the intervention and the active management policy (please see S3) and we have further made a reference to the RCT-article (1). 

Reviewer #1, comment 2

2. The aim of the study as it reads is to obtain manager perception of sickness absence while in the introduction there is more reference to perspectives of pregnant staff. Statements such as studies 'show that job adjustments are only made for half of the pregnant employees who state a need for it. This could indicate lack of focus on the working environment among pregnant women' makes it confusing if pregnant staff are not seeking their entitlements or if the managers are denying it.

Author’s response to the Reviewer #1, comment 2

Thank you for this interesting angle. We are aware that the introduction, to a large extent have more references to the perspectives of the pregnant women and not the manager. There is a lack of research concerning the manager’s perspective and therefore a lack of references. We agree that there is a source of tension here, which can be difficult to navigate in as a manager (and as a pregnant employee). With this study, we hope to shed some light on this from the manager’s perspective. Please also see reviewer 3, comment 4. We have added a sentence to clarify this.

Page 4, line 80 

There is a lack of studies identifying these issues, from a managerial point of view.

Reviewer #1, comment 3

3. The methods section needs to be clarified further as in one instance it is said that there were 5 FGDs but there were 19 participants in total. Is it possible that some of the FGDs had 4 or less participants? why were there 5 FGDs in the first place? and were there any differences in who participated in which FGD (this is mentioned in limitations but need to be explained upfront in the methods section itself).

Author’s response to the Reviewer #1, comment 3

Thank you for giving us the opportunity to clarify this further. The intervention in the RCT? was performed over a large geographic area ranging from Copenhagen to the Zealand region, which in total includes five hospitals. Recruitment was carried out in order to ensure information rich cases, with as much variation as possible. However it was also kept in mind not to create too much inconvenience for the managers to be able to participate. Therefore 5 focus groups interviews were planned and carried out, one in each hospital. Initially, the target number of informants for each group was 5-6, however we were challenged by a lot of last minutes cancellation from the managers, due to busyness. 

There were no differences in who participated in the focus group interviews, they were all managers. The same interview guide was used in all interviews (please see the interview guide in the supporting information S2) meaning that the topics discussed were the same.

Changes was made at page 5 and 6

To clarify these questions, the description of the participants and their enrollment in the focus group interviews are now under “Procedure and sample”. We have also added a sentence to explain why five focus groups were undertaken.

Page 6, line 126 

Five focus groups interviews were planned, one in each of the five participating hospitals. This was done to ensure that it would be convenient for the managers to participate.

Reviewer #1, comment 4

4. Given that the FGDs were held a year after the seminar participation of the managers, it is possible that all their views are after learning the correct approaches and can be a major limitation of the study.

Author’s response to the Reviewer #1, comment 4

Thank you for this interesting comment. The average years of experience for the managers in this study was 12.6 years, meaning that the majority had years of experience before the seminar. Furthermore, the interview guide was structured with open and broad initial questions starting with general experiences and later narrowed down to questions related specifically to the seminar and the active pregnancy policy (please see the interview guide). We believe that these factors made it possible for the managers to distinguish between the general experiences and specific experiences of using active management policy. During the focus group interviews, the managers also often used terms such as; ‘before the seminar’ and ‘now’ referring to two different periods. From this, we have no reason to think that taking part in the seminar changed the mangers general experiences significantly. 

Reviewer #1, comment 5

5. The chosen themes are not self-explanatory and do not seem corroborating with the sub-themes. Themes are usually more crisp and punchier than the sub-themes. The articulation of themes need to be from the manager perspective - as it seems it is unclear if they are manager perspectives of staff perspectives.

Author’s response to the Reviewer #1, comment 5

Thank you for this comment and the opportunity to reconsider and adjust the themes more toward the managers’ perspective. We went through the themes and sub-themes and made the themes clearer and more corroborating with the sub-themes. Changes have been made throughout the manuscript accordingly.

Page 9 Table 2 

Changed from – to:

Work retention and everyday management - The everyday management

Dissimilarity among pregnant staff members - Managerial dilemmas

The dynamics of the workplace culture - Acknowledging workplace culture

Reviewer #1, comment 6

6. There can be a better choice of quotes to align with the themes - it is important to mention if the sub-themes were coming from many participants or just one participant in one FGD.

Author’s response to the Reviewer #1, comment 6

Thank you for this interesting comment. Firstly, our intention with the use of quotations in the manuscript was to use them as examples, to illustrate only some of the content of our findings. Furthermore, the quotations were used to bring the text to life by Malterud (2) and not with the purpose of validating our findings as recommended 

We have gone through all the quotations again and have replaced two quotations 

Page 10, line 215 replaced

“They come to me almost with the positive test in hand….It is really good because it’s helpful also for the staff member. While I’m planning for a long period of time, I’m actually able to help them a good while ahead. Spread out their night shifts, making sure their shifts aren’t too long.” (FG 2)

With:

“You stay ahead of things and align your expectations and get that signal sent; you know what, I will do everything possible so that you can stay at work, there are plenty of options” (FG 2)

Page 15, line 334 replaced 

“This thing, where instead of sitting down and telling them to remember to look after themselves, some people say: pregnancy is not an illness, it’s great, remember to enjoy it, yes all that’s a part of having a child at the end of it.” (FG 1)

With

“…if there is a culture that dictates that when you are pregnant, you have to take sick leave, then I believe the attitude will spread, and more will eventually become sick listed” (FG 3)

Secondly, for our data analysis we used thematic analysis as described by Braun and Clarke (3) This method seek to find and describe repeated patterns of meaning across a dataset. All Themes and sub-themes are identified based on multiple meaningful units (codes) across the dataset. This may not have been clear in the manuscript and we have added a sentence to make this more clear. 

Page 8, line 168 

Themes and sub-themes were identified based on multiple meaningful units

Reviewer #1, comment 7

7. Given that the study takes an inductive perspective, the themes need to be brought together to develop a coherent theory and this is not clear in the discussion.

Author’s response to the Reviewer #1, comment 7

Thank you for this comment. Thematic analysis used in the present study described by Braun and Clarke (3), is about understanding peoples’ everyday experiences of reality, in great detail, so as to gain an understanding of the phenomenon in question. It does not necessarily include the development of a coherent theory as e.g. grounded theory does. We have therefore not added further in the discussion 

Reviewer #1, comment 8

8. Reflexivity and positionality of the authors need to be made more explicit in the methods section.

Author’s response to the Reviewer #1, comment 8

Thank you for this comment. We agree that reflexivity and positionality is not explained thoroughly enough in the manuscript and we have therefore added the following in the text. 

Page 5, line 115

With the aim of the study, we seek to understand the basis on which managers make their decisions, when they interact with pregnant employees. We acknowledge that we are part of the analysis and that we consciously use all members of the research team supplement and contest each other’s statements during the analysis (2). 

Reviewer #2, comment 1

I congratulate the authors to an interesting paper. I have some comments.

Author’s response to the Reviewer #2, comment 1

Thank you very much; we hope that the manuscript will add new knowledge to the field and to shed some light on the complexity of the topic.

Reviewer #2, comment 2

Triangulation mostly describes involving using multiple sources of data in an investigation. The authors describe that they used triangulation meaning that different health care professionals/researchers were analyzing the data. However, this study could be referred to a mixed method study if the qualitative study was planned from the project’s start to complement the RCT. But, the published study protocol does not describe the qualitative study, eg population, data collection or the qualitative analysis.

Author’s response to the Reviewer #2, comment 2

Thank you for this comment. It is true that we used researcher triangulation and could have planned the study as a mixed method study, which would have been an interesting approach. The present study was, however planned after commencement of the RCT study.

Reviewer #2, comment 3

I want more information about the intervention ie. the education of the leaders, which can increase the value of the study’ and make it reproducible.

Author’s response to the Reviewer #2, comment 3

We appreciate this comment and have provided an additional file, with thorough information about the intervention as a hole and the active management policy concept specifically.

Please see supporting information S3

Reviewer #3, comment 1

ABSTRACT

It is unclear from reading the abstract by itself where this study took place. An additional sentence clarifying location and sample population would be useful.

Author’s response to the Reviewer #3, comment 1

 Thank you for this comment. We have added a sentence in the abstract to clarify this.

Page 2, line 34. Changed from:

A qualitative study based on five focus group interviews (N=19) was conducted with participation of hospital managers from 17 different wards.

To:

A qualitative study based on five focus group interviews was conducted at five public hospitals in Zealand, Denmark with participation of 19 hospital managers from 17 different wards, representing six different medical specialties.

Reviewer #3, comment 2

DATA AVAILABILITY

Per my reading of the PLOS ONE guidelines, my understanding is that the Methods section should contain information regarding how to access the data used in this study - even if the full dataset is not publicly available.

Author’s response to the Reviewer #3, comment 2

Thank you for this comment. We have added information on data availability in the methods section.

Page 8, line 184

Modified data is available upon request

Reviewer #3, comment 3

PREDICTIONS

Even though this study is largely qualitative in nature, a set of predictions would be a good addition

Author’s response to the Reviewer #3, comment 3

We agree that the current study has a connection with a RCT study, however we believe that the nature of the present study is fully qualitative and we have therefore decided not to add predictions. Please see reviewer 1, comment 8, on Reflexivity and positionality. We have added a sentence.

page 5, line 115:

With the aim of the study, we seek to understand the basis on which managers make their decisions on, when they interact with pregnant employees. We acknowledge that we are part of the analysis and that we consciously use all members of the research team supplement and contest each others’ statements during the analysis (2). 

Reviewer #3, comment 4

DISCUSSION

Much of this paper hinges on the idea of unnecessary sick leave (at least from the manager perspective) and how to prevent it. I would've liked to see a deeper discussion of pregnancy itself and how variable symptom prevalence/severity, incidence of medical complications, and real occupational threats factor in to the dynamic between pregnant individuals and the workplace. 

How does one determine if sick leave is necessary or not? I think this is especially critical, given the observation that medical practitioners appear to have a very different threshold for workplace risk than managers. 

Author’s response to the Reviewer #3, comment 4

Thank you for this very interesting comment. It did became clear though the interviews that managers have a different perspective on sick leave during pregnancy. Managers often do not have the health professional credentials to assess whether sick leave is necessary or not. Furthermore, pregnancy-related discomforts are often the course of sick leave and only the pregnant employee are able to determine when she can or cannot work. This field is complex, which the interviews also reflected.

It is well known that pregnancy may course discomforts such as low back pain, which can make it difficult to work in a busy hospital ward. It is the responsibility of the employer not only to adjust job tasks in order to ensure that employees are not exposed to occupational risks, but also to ensure that pregnant employees are able to work, without getting overloaded. Unnecessary worry may be defined as fear of factors being harmful to the fetus, when there is evidence that this is not the case. Information and knowledge given to the pregnant employee, may decrease this worry or fear and thereby prevent unnecessary sick leave. 

We have added a sentence in the discussion

Page 20, line 448

Managers often do not have the health professional credentials to assess whether sick leave is necessary or not, however they may have knowledge to overcome unnecessary worry and thereby prevent unnecessary sick leave. 

Reviewer #3, comment 5

I'd also like to know more regarding actual exposure risk in these work environments and to what degree women are right to be worried about possible infection (especially considering certain aspects of immune function are down-regulated during pregnancy, producing increased risk of viral infections). 

Author’s response to the Reviewer #3, comment 5

Thank you for this comment. The intervention was developed specific with the purpose of addressing potential worries and occupational risks among pregnant staff members and to accommodate these by relocating pregnant staff members to other patients or tasks in cases of risk of infection. During the development of the intervention, a hygiene specialist was consulted and stated that all hospital staff members are encouraged to comply with Hospital standards regarding hygienic precautions and use protective equipment as recommended in each patient category. 

This study was conducted from February to May 2019, before the Corona pandemic became a reality. We have added text in the discussion to address this.

Page 21, line 465

According to hygiene specialists, all hospital staff members are encouraged to comply with Hospital standards regarding hygienic precautions and use protective equipment as recommended in each patient category. This is considered sufficient to avoid infectious diseases.

Reviewer #3, comment 6

How many of these managers oversaw acute health settings (e.g. ICU) versus more chronic long-term care (e.g. med-surg), and does this factor in to employee anxiety regarding sick leave? 

Lines 171-175 give some idea of the dataset makeup, but little is said regarding possible differences between types of healthcare in the results/discussion.

Author’s response to the Reviewer #3, comment 6

Thank you for this interesting perspective. The participants came from various fields; medical and surgical departments, pediatrics, gynecology and obstetrics, emergency medicine, and neurology departments. Several of the wards have emergency patients, however, the ICU was not represented among participants. The aim of this, was to cover different perspectives and issues across the medical specialties. The data did not indicate that anxiety or fear of getting an infection was greater at some wards than others. 

Reviewer #3, comment 7

LINE EDITS

line 72-73: The wording here is confusing. Do pregnant women themselves lack awareness, or do their managers/co-workers lack awareness? I assume you mean the latter, but restructuring this sentence would easily clear this up.

line 238: Unnecessary use of "e.g."

line 264: Labeling all concerns from friends and family as "myths" is probably too strong a statement. The way it is currently written does not clarify whether this was the chosen wording of the managers or of the authors.

Author’s response to the Reviewer #3, comment 7

Thank you for the line edits. They have all been corrected according to the recommendations. Please see in the manuscript.

1. Begtrup LM, Malmros P, Brauer C, Soegaard Toettenborg S, Flachs EM, Hammer PEC, m.fl. Manager-oriented intervention to reduce absence among pregnant employees in the healthcare and daycare sector: a cluster randomised trial. Occup Environ Med. 12. januar 2021; 

2. Malterud K. Qualitative research: standards, challenges, and guidelines. Lancet Lond Engl. 11. august 2001;358(9280):483–8. 

3. Braun V, Clarke V. Using thematic analysis in psychology. Qual Res Psychol. 1. januar 2006;3(2):77–101.

---

## [Editor Report · Decision Letter 1]

10 Feb 2021

Experiences managing pregnant hospital staff members using an active management policy – A qualitative study

PONE-D-20-20150R1

Dear Ms. Backhausen,

We’re pleased to inform you that your manuscript has been judged scientifically suitable for publication and will be formally accepted for publication once it meets all outstanding technical requirements.

Kind regards,

Sathyanarayanan Doraiswamy, MD, DHealth

Guest Editor

PLOS ONE

Additional Editor Comments (optional):

The authors have done a very good job in factoring in the reviewer comments. This updated version is rich in content and has a nice flow to it. Well done!

In the abstract you say 'Four thematic themes' under Results - you may want to edit it to just say 'themes'.
---

## [Editor Report · Acceptance letter]

17 Feb 2021

PONE-D-20-20150R1 

Experiences managing pregnant hospital staff members using an active management policy – A qualitative study 

Dear Dr. Backhausen:

I'm pleased to inform you that your manuscript has been deemed suitable for publication in PLOS ONE. Congratulations! Your manuscript is now with our production department. 

Kind regards, 

on behalf of

Dr. Sathyanarayanan Doraiswamy 

Guest Editor

PLOS ONE